# Application of a Cold-Pressing Treatment to Improve Virgin Olive Oil Production and the Antioxidant Phenolic Profile of Its by-Products

**DOI:** 10.3390/antiox12061162

**Published:** 2023-05-27

**Authors:** África Fernández-Prior, Juan Cubero Cardoso, Alejandra Bermúdez-Oria, Ángeles Trujillo Reyes, Juan Fernández-Bolaños, Guillermo Rodríguez-Gutiérrez

**Affiliations:** Instituto de la Grasa, Consejo Superior de Investigaciones Científicas (CSIC), Campus Universitario Pablo de Olavide, Edificio 46, Ctra. de Utrera, km. 1, 41013 Seville, Spain; mafprior@ig.csic.es (Á.F.-P.); juan.cubero@ig.csic.es (J.C.C.); alebero-ri@ig.csic.es (A.B.-O.); atrujillo@ig.csic.es (Á.T.R.); j.fb.g@csic.es (J.F.-B.)

**Keywords:** phenolic, hydroxytyrosol, 3,4-dihydroxyphenylglycol, cold pressure, olive oil solid waste, antioxidants, by-products

## Abstract

The olive oil sector is continuously evolving in order to improve the quality of olive oil and its by-products. In fact, the trend is to use increasingly greener olives to improve quality by decreasing the extraction yield, thus obtaining a higher content of antioxidant phenolics. The application of a cold-pressing system to the olive before the extraction of oil was tested with three varieties: picual at three different stages of maturity and arbequina and hojiblanca at early stages of maturity. The Abencor system was used for the extraction of virgin olive oil and its by-products. For the quantification of phenols and total sugars for all phases, organic solvent extractions and colorimetric measurements and high-performance liquid chromatography (HPLC) with a UV detector were used. The results show that the new treatment significantly improved the amount of oil extracted by between 1 and 2% and even increased its concentration of total phenols by up to 33%. Regarding the by-products, the concentrations of the main phenols, such as hydroxytyrosol, increased by almost 50%, as did the glycoside. The treatment also facilitated the separation of phases in by-products and improved the phenolic profile, although not in terms of total phenols, but individual phenols with higher antioxidant activity were obtained.

## 1. Introduction

Scientific advances in the food area are focused on developing new technologies for increasingly safer and healthier foods while promoting the integral use of agroindustrial by-products by obtaining high-value-added components [1]. The olive oil sector is a clear example of this evolution. The main by-product generated in this industry in Spain is so-called alperujo, which is a solid with high moisture (60–70%) generated from the continuous two-phase olive oil extraction system. New pre-treatment systems, mainly thermal ones [2], are being implemented in olive pomace extractors that allow the extraction of some of the bioactive compounds, improving the possibilities of application to the rest of the components. In fact, extracts rich in one of the main active phenols from olives, hydroxytyrosol, are currently being commercialized [3], and others, such as extracts rich in 3,4-dihydroxyphenylglycol, are on the way [4]. The potential of these bioactive compounds for use in different fields, such as food, pharmaceuticals or cosmetics, is very high, and high demand for these natural products with antioxidant, anti-inflammatory, antitumor, etc., activities is expected [5]. Researchers have always sought to increase the concentration of phenols in olive oil, since they are among the compounds responsible for healthy properties [6]. Systems that improve heat transfer in the beating and the vacuum in the beating phase, along with other techniques, such as ultrasound and microwave-assisted pre-treatment, have been used to improve the properties of the oil and its extraction [7,8]. However, in addition to these factors, in order to achieve sustainability in the olive oil production chain, it is also necessary to develop pre-treatments that improve the management of by-products. The application of a vacuum improves oil extraction and reduces oxidative processes but does not promote the formation of the compounds involved in the aroma of the oil itself, reducing its organoleptic characteristics. On the other hand, emerging techniques such as ultrasound, which increases the concentrations of certain compounds, such as tocopherols, chlorophylls and carotenoids, or microwave treatment [9] increase oil extraction without substantially altering its quality parameters.

These techniques can help to improve and simplify the oil extraction system, allowing a better fat yield in the case of greener olives, but their impact on the by-products generated has not been studied. At the same time, their energy consumption must be taken into consideration. The aim of the application of new technologies is to facilitate cell rupture to release microdroplets from the oil while improving the contact between the oil and minor compounds, facilitating their solubilization and promoting reactions for the formation of volatiles and more liposoluble phenols. Recently, a new cold-pressing technology has been developed to improve fresh food preservation.

In this type of treatment, very high pressures are used for short periods of time (600 MPa and 2–5 min). For the treatment of olives prior to oil extraction, a high-pressure system could initiate cellular rupture, facilitating the subsequent malaxation phase and increasing oil extraction and the contact between the oil and vegetation water and all the functional components of the olive. However, the use of pressure levels developed for the microbial decontamination of foodstuffs can be excessive in promoting the rupture of the fruit, producing undesirable reactions such as hydrolysis or fermentation, and can also involve excessive investment and operating costs. For this reason, a new treatment using pressures 1000 times lower that is more easily adaptable to the olive oil industry and would represent an affordable investment is being tested. In the present work, the use of pressures between 0.39 and 0.69 MPa and a residence time of 10 min, which could be easily applied in the industry, was studied. We focused on improving the extraction process with green olives and facilitating the subsequent management of its by-products, improving the phase separation and the solubilization of the main phenols in the aqueous phase. For this purpose, different varieties of olives with low maturity indices were studied by evaluating the fat yield and the solubilization of the main functional phenols in the oil, vegetation water and alperujo (Figure 1).

## 2. Materials and Methods

### 2.1. Samples

Different batches of olive samples were treated during the experiment. At the beginning of the 2020/2021 season, olive samples from three varieties were collected from the Instituto de la Grasa field collection, Seville, Spain: picual, arbequina and hojiblanca. In the middle of the 2020/2021 season, picual olive samples with a higher maturation index were collected from the Pedroches Valley, Córdoba, Spain. Finally, in the middle of the 2021/2022 season, hojiblanca olive samples with a low maturation index were collected in Granada.

### 2.2. Standard Compounds

For this work, 3,4-dihydroxyphenylglycol, syringic acid, verbascoside and luteonin-7-O-glucoside were obtained from Sigma-Aldrich (Deisenhofer, Germany). Tyrosol was obtained from Fluka (Buchs, Switzerland) and hydroxytyrosol was obtained from Extrasynthese (Lyon Nord, Geney, France). Gallic acid, trifluoroacetic acid, anthrone, glucose, Folin–Ciocâlteu phenol reagent, vanillic acid, oleocanthal, oleacein and 2,2-diphenyl-1-picrylhydrazyl were obtained from Sigma-Aldrich (Madrid, Spain). Sodium bicarbonate (Na_2_CO_3_), acetonitrile (HPLC grade), hexane and sulfuric acid (H_2_SO_4_) were purchased from Panreac Quimica S.A. (Barcelona, Spain).

### 2.3. Cold-Pressure Treatment

A stainless-steel reactor with 100 L capacity was used at the experimental plants of the Instituto de la Grasa-CSIC. The olive samples were placed in the hermetically sealed reactor. Subsequently, pressurized air was introduced until the desired pressure was reached (0.39 and 0.69 MBa), after which the treatment was maintained for up to 10 min. Once finished, the sample was slowly depressurized through a valve before being removed from the reactor. The temperature of the samples did not exceed 0.2 °C.

### 2.4. Olive Oil Extraction

The oil was extracted from the treated olives and from the control olives without cold treatment using an Abencor system (MC2 Ingenierıa Sistemas, Seville, Spain). The olives were initially milled with a hammer mill with a 4 mm outlet filter. After that, a malaxation system was applied using three jars with 600 g per sample at a temperature of 29 °C for 45 min. During this time, 100 mL of water was added to each jar. Another three jars of 600 g were used for the control. Six jars were used for each sample. After malaxation, the samples were centrifuged into separate two phases: the liquid and the solid (or alperujo) phases. From the liquids, the aqueous and oily phases were separated into one-liter flasks and the amount of oil and water extracted for each sample was measured. The samples were stored at −20 °C until analysis.

### 2.5. Determination of Moisture Content

The alperujo moisture content was determined following the recommendations of the Standard Methods of the American Public Health Association [10]. Approximately 2.5 g of sample was dried in a laboratory stove (J.P. Selecta) at 105 °C for 24 h until samples reached constant weight.

### 2.6. Determination of the Maturity Index

The maturity index (MI) was determined following the method described by Uceda and Frías [11]. A sample of 1 kg of olives was homogenized. Then, 50 fruits were randomly separated and visually classified according to eight categories ranging from intense green to black. Finally, the following formula was applied:IM = (A × 0 + B × 1 + C × 2 + D × 3 + E × 4 + F × 5 + G × 6+ H × 7)/100
where A, B, C, D, E, F, G and H are the numbers of fruits in each category (0, 1, 2, 3, 4, 5, 6 and 7).

### 2.7. Extraction and Determination of Total Phenolics and Sugars

The oil samples were extracted following the method described by Vázquez-Roncero et al. [12]. Briefly, 10 g of oil was weighed into a flask and dissolved in 50 mL of hexane. Three liquid–liquid extractions were employed using a separating funnel with 20 mL of a methanol:water (60:40 (*v*/*v*)) mixture, shaking it each time for 2.5 min. Finally, the aqueous fraction was separated and stored at −20 °C until analysis.

The alperujo samples were extracted following the method developed in previous work (Fernández-Prior et al., 2020). First, 10 g of alperujo was extracted three times with 20 mL of methanol:water (80:20 *v*/*v*). The mixtures were subsequently triturated in an Ultra Turrax IKA T25 digital blender for 60 s at 1000 rpm. The liquid phase was separated with centrifugation for 20 min at 5800 g in a Sorvall RT 6000 D centrifuge. Finally, the methanol was evaporated in a vacuum at 40 °C and the samples were stored in water at −20 °C until analysis.

The water samples were centrifuged at 12,100 g for 14 min and then filtered through paper (0.45 µm).

Total phenolics were determined using the colorimetric Folin–Ciocâlteu method following the procedure described by Obied et al. [13]. First, 80 µL of Na_2_CO_3_ at 0.7 M was added to 20 µL of each extract. Then, 100 µL of the 0.2 M Folin solution was added to each plate well. After 15 min, the absorbance of the samples was recorded at 655 nm in a Bio-Rad iMark model microplate reader (Hercules, CA, USA). Finally, a calibration curve with a known gallic acid concentration was developed, and the results were expressed as milligrams of gallic acid equivalents per kg of oil, per L of water or per kg of alperujo.

The Anthrone colorimetric method was used to determine the total sugar contents [14]. Then, the absorbance of the samples was recorded at 630 nm in a Bio-Rad iMark model microplate reader (Hercules, CA, USA). Finally, a calibration curve with a known glucose concentration was developed, and the results were expressed as milligrams of glucose equivalents per kg of alperujo or per L of water.

### 2.8. Analysis of Individual Phenolics with High-Performance Liquid Chromatography (HPLC)

Identification and quantification of the main individual phenolic compounds were performed with a Hewlett-Packard 1100 series high-performance liquid chromatography (HPLC) system, (Agilent, Barcelona, Spain). The HPLC system was equipped with a diode array detector (DAD) and an Agilent 1100 automatic injector (sample injection loop of 20 µL). The chromatographic column used was a Teknokroma Tracer Extrasil OSD2 with a 5 µm particle size and 25 × 0.46 mm internal diameter. The mobile phases were Milli-Q water (pH adjusted with 0.01% trifluoroacetic acid) (solvent A) and acetonitrile (HPLC-grade) (solvent B). The elution method consisted of the following gradient over a total run time of 55 min: 0–30 min, 5% B; 30–45 min, 25% B; 45–47 min, 50% B; 47–50 min, 0% B, which was maintained until completing the run. The identification and quantification were undertaken using commercial standards by comparing the retention times with that of the reference compound and recording the UV spectra in the 200–360 nm range. The results were expressed as mg of each phenolic per kg of alperujo or per L of water.

### 2.9. Statistical Analysis

All results were expressed as the mean values ± standard deviation from replicate experiments. STATGRAPHICS^®^ Plus software (Statgraphics-net, Madrid, Spain) was used for statistical analyses. Comparisons between samples were performed using one-way analysis of variance (ANOVA) and the least significant difference (LSD) method at the same confidence level. A significance level alpha or *p*-value of <0.05 indicated statistically significant differences.

## 3. Results

### 3.1. Oil and Water Extractability and Moisture

The samples studied were of the picual, arbequina and hojiblanca varieties, which are the most commonly used for the extraction of olive oil in Spain. These three varieties were studied with low maturity indices, which made them green, as they have strong texture and oil extraction yields are lower. However, greener or less ripe olives are increasingly used to obtain quality oil. Subsequently, the study was extended with two higher-maturity indices of 1.2 and 2.4 for the picual variety only, as it is the most representative of the sector. Figure 2 shows the values for the maturity indices, the moisture content in the olive samples and the quantities of olive oil and aqueous fraction extracted with the Abencor extraction system. The cold-pressing treatment did not significantly modify the moisture. The amount of oil extracted in the case of the arbequina samples increased significantly by about 1%; in the case of hojiblanca, there were no significant differences. Among the picual samples, for the less mature samples, the grade yield decreased in the treatment at 0.39 MPa but increased by 2.5% when the pressure was 0.69 mPa. This result was reproduced again with the picual variety at the two higher maturity stages of 1.2 and 2.4, increasing the oil recovery by 2.3 and 1.9%, respectively.

The extractability of the aqueous phase did not vary in the arbequina variety with an MI of 0.8 and decreased in the hojiblanca variety with an MI of 1.1, reaching 1.9 and 2.9% for the minimum and maximum pressures, respectively. In the case of the picual variety, the behavior was different since, for the MI of 0.9, the use of a pressure of 0.39 MPa improved water extraction by 4.1%, while using the highest pressure significantly decreased it by 3.4%, and no differences were found for the highest maturity stage.

The hojiblanca samples collected in the last season showed lower moisture, which implied very low recovery of the liquid fraction. Nevertheless, the oil recovery was very similar to that of the same variety in the previous season. The cold-pressure treatment was not effective for this sample for which the moisture was very low.

### 3.2. Alperujo Characterization

The alperujo, or semi-solid residue, obtained from the Abencor extraction system was similar to the by-products obtained at the industrial level in two-phase extraction systems, so the results shown in Table 1 are significant for the evaluation of the effect of cold-pressing treatment on the release of phenols in alperujo at the industrial level. The table shows the soluble sugars content, as well as the total and individual soluble phenols content. The soluble sugars content increased in the arbequina variety only when the pressure of 0.39 MPa was used, while for hojiblanca, it increased in the same way with the two pressures, and no significant variation was observed in the picual samples for the three stages of maturity.

In the case of total phenols, the behavior was similar to that of sugars with the exception of the arbequina variety, for which no significant variation was observed, as with the picual variety for the three stages of maturity. An increase in total phenols was observed for the two pressures used with the hojiblanca variety, as was the case for sugars.

In the case of individual phenols, the behavior was not generalizable but depended on each type of phenol. Thus, 3,4-dihydroxyphenylglycol (DHPG) showed a tendency to decrease with the use of cold pressing. This decrease was not significant for the arbequina variety, but very pronounced decreases were observed for both hojiblanca and picual, with the exception of the 3.9 MPa treatment with an MI of 1.2. Hydroxytyrosol 4-β-D-glucoside (HT-Glu) is one of the precursors of hydroxytyrosol (HT) and originates from the partial hydrolysis of oleuropein, so its presence indicates that the samples are fresh and the transition to HT has not been completed. HT-Glu significantly increased in concentration in most cases, reaching an increase of almost twofold in the hojiblanca variety. Similar behavior was presented by HT, which increased in most cases but remained the same in some cases, such as in picual in the more mature and less mature stages. The behavior of tyrosol (Ty) was very similar to that of HT except for the arbequina variety, for which a decrease was observed for the use of 3.9 MPa. For the rest, increases were observed when using the two pressures in the case of hojiblanca, and in picual, the tyrosol concentration was maintained in the samples with MIs of 0.9 and 2.4 and increased in that with an MI of 1.2. Syringic acid (Sy A) clearly increased in the arbequina and hojiblanca samples. In the picual variety, it did not appear in the greenest samples but increased for the treatment at 0.39 MPa. Verbascoside showed a rise in concentration in arbequina for the 0.39 MPa treatment, and it also increased for the hojiblanca variety in the two treatments. In the case of the picual samples, it did not appear in the least mature sample and decreased in the others. Luteonin-7-O-glucoside (Lu-Glu) increased in all varieties and for the two cold-pressure treatments except in the greenest picual sample, for which only traces appeared. Finally, comsegoloside increased with pressure in general, especially with the use of 3.9 MPa for all varieties.

The hojiblanca samples collected in the last season showed a lower concentration of total sugars, while their total phenol content was higher than the rest of the samples. The high concentrations found for comsegoloside and luteonin glucoside, which were the only phenols for which the treatment increased their concentration, were noteworthy.

### 3.3. Characterization of the Vegetation Water

The amount of water extracted with the Abencor system is not representative of separation in a two-phase system, but the amounts and concentrations of sugars and phenols give an indication of the effect of the cold-pressing treatment on the by-products and whether the pressure treatment itself could improve the subsequent phase separation in the alperujo. Table 2 shows the sugars and total soluble phenols content in the methanol:water extraction, as well as the main individual phenols found with HPLC-UV.

There were no significant differences in total sugar values for any of the varieties except for the picual variety at a pressure of 0.39 MPa, for which, despite the significant increase, the value was not very high. In the case of total phenols, there was no variation. However, there were differences in the phenolic profiles. DHPG increased in all cases, especially after the use of the higher pressure of 3.9 MPa, with the exception of the Arbequina variety, for which no variation was observed. The concentration of HT-Glu decreased slightly in all cases except for the picual variety in its less mature stage, for which the concentration increased significantly and decreased with an MI of 1.2. The same behavior was observed for HT and its glycoside for the arbequina and hojiblanca varieties, while the concentration increased in the case of picual with MIs of 0.9 and 1.1 and decreased for the more mature stage. The concentration of tyrosol did not change much, only decreasing drastically in the arbequina variety at the maximum pressure studied and increasing significantly in the picual variety with an MI of 0.9. Syringic acid showed an increasing trend in all cold-pressing treatments, with only the increase in the hojiblanca variety being significant. Verbascoside maintained its concentration in all varieties except picual. In the cases of both comsegoloside and p-Co, no significant variations were observed in any of the different varieties or stages of maturity when cold-pressing treatments were applied.

The behavior of the hojiblanca sample from the following season was different due to its low moisture and higher phenol concentration. The high comsegoloside content was noteworthy and the treatment did not influence the concentrations of these phenols.

### 3.4. Phenolic Characterization of the Extracted Olive Oil

The total and individual phenols in the oil samples extracted with the Abencor system were determined for each of the olive samples (Table 3). The arbequina samples showed the lowest concentration of total phenols, with a 25% decrease in the oil with the use of cold-pressing treatments. In the case of hojiblanca, it decreased by 33% with the lowest pressure but increased by 11% with the use of the 3.9 MPa treatment. In the case of the picual variety, the results were not homogeneous and depended on the state of maturity, increasing in the case of treatment at 0.39 MPa for the greenest olives and when the highest pressure was used with the ripest samples. For the individual phenols, in general, small decreases or increases were observed that did not noticeably change the profile in the oil. DHPG, the concentration of which was essentially maintained with the use of the higher-pressure treatments, showed the highest increase with the higher pressure in the more mature picual samples. The behaviors of HT and Ty were very similar. Their concentrations decreased slightly in all cases except for with the lowest pressure in the greenest picual samples. The secoiridoid derivatives oleacein and oleocanthal were present in greater quantities in olive oil and represent two of the most important phenols derived from oleuropein and ligustroside. In this case, there were no major variations, and both slightly decreased in the arbequina and hojiblanca varieties and with almost all the treatments carried out with the picual variety. The hojiblanca variety from the last season showed the highest concentration of total phenolics.

## 4. Discussion

The use of a cold-pressing system with olives prior to olive oil extraction was evaluated. The study focused on olives with a low state of maturity, looking for higher-quality oil in contrast to oil yield. The cold-press system was studied in an attempt to improve oil extraction and determine whether the solubilization of phenols could be improved in each of the fractions derived from the oil extraction, such as olive oil, vegetation water and alperujo. Three of the most widely used varieties for obtaining olive oil in Spain—namely, picual, arbequina and hojiblanca—were chosen for the study. Two pressure conditions easily scalable at the industrial level were tested (0.39 and 0.69 MPa) with the three varieties with similar states of maturity. Finally, the study was extended by testing two higher-maturity stages for the picual variety, which was the most representative as it is the most widely used in the Spanish sector, and a hojiblanca sample harvested from the following season.

### 4.1. Effect of the Treatment on Oil Extractability and Water Separation

The application of the cold-pressing treatment did not lead to a significant increase in temperature in any of the cases (data not shown), with an average temperature for the samples before treatment of 12.2 °C and an average after treatment of 12.3 °C. The data for oil yield and vegetation water extraction showed that the treatment caused certain structural changes that facilitated these extractions. Although these results were not homogeneous, it can be affirmed that there was a clear and significant trend towards an increase in oil yield in two of the three varieties studied, arbequina and picual, where it increased by over 1% and 2%, respectively. These percentages are very significant and are higher than those found by other authors using more expensive technologies—such as ultrasound, which, in the best of cases, improved the percentage of extracted fat by 0.4 points [14]—and those reported by Clodoveo and Hbaibed [7] for the use of microwaves or ultrasound, which, in the best of cases, increased the oil extraction percentage by 0.4 points. This is why this system could lead to a significant improvement in the extraction of oil from green olives or olives with a low degree of maturity at the industrial level.

### 4.2. Effect of Treatment on Alperujo

The data obtained for alperujo showed that, in the arbequina and hojiblanca samples, the sugars content increased, while in picual samples, it did not change. In other words, in none of the cases did the sugars concentration decrease, and the possible improvement in the solubilization of sugars implied that the hydrolysis mechanisms were activated to a greater extent after the use of the cold-pressing system. The advantages focus on the fact that it may be easier to obtain extracts richer in sugars after the application of the thermal processes that are implemented in the sector for alperujo utilization [2], and it may even be possible to improve the application of bioprocesses for complete utilization [15].

In the case of total phenols, there were practically no differences between applying the treatment and not applying it, but there were differences with respect to the phenolic profile. DHPG and comsegoloside hardly changed in concentration after treatment. However, hydroxytyrosol and its glycoside, which is also one of its precursors, increased markedly, as did other interesting phenolics, such as verbascoside and luteonin-7-O-glucoside. This is very important because HT is one of the most active phenols and one of the most abundant in olives and their by-products [3,16]. It is also the most abundant phenol in free form and the one that is making it possible to implement true biorefining of olive pomace, beginning with the extraction of its bioactive compounds. Extracts rich in HT are already being commercialized [17] from olive by-products. The amounts of HT obtained with the application of the cold-pressing system, which are greater than 1000 ppm, are very similar to those obtained after the application of one of the thermal systems used in the industry, such as 60 °C for 60 min, although they are still far from the 5000 ppm obtained after the application of more severe industrial treatments, such as 160 °C for one hour [18]. In the case of alperujo, the use of a three-phase system, which facilitates obtaining an aqueous source rich in phenols for the their extraction in pomace extractors, is being promoted. The use of this system is increasing [2], but it has the drawback of requiring long storage periods for the liquid source in order to promote the hydrolysis of the HT precursors for release. The results shown in this work verify that the cold-pressing system can already initiate hydrolysis reactions with marked increases in the concentrations of HT and its glycoside, which can help to decrease storage time and even increase the content of this phenol in the liquid fraction.

### 4.3. Effect of Treatment on the Water Fraction

Studying the extraction of this water can provide information on the solubilization of phenols into the aqueous phase during the malaxation and milling phases. Therefore, characterization of this phase can indicate whether phase separation in the alperujo improved and how the main bioactive compounds, the phenols, behaved after the application of the new cold-pressing treatment for the subsequent use of the by-product in a sustainable process [19].

There were no significant differences with respect to the concentration of sugars present in the aqueous fraction. Unlike most of the main simple phenols, the precursors of low-polymerizing sugars were more difficult to hydrolyze as they belong to the wall material and mainly the hemicellulosic fraction [20]. Therefore, they require more severe treatment for their hydrolysis and for the solubilization of sugars or longer reaction times, which is why no differences were observed with the use of the pressurized system. In the case of individual phenols, differences were seen in that they promoted chemical and enzymatic reactions more quickly than other phenols that were already soluble.

The fact that the total phenolic content did not increase means that the extraction of phenols, which may have been bound to the structure of the plant wall, was not facilitated, but the profile was changed due to the promotion of reactions in the soluble phenols, mainly from the secoiridoids, such as oleuropein and ligustroside, which are the most abundant in olives [21,22]. It is for this reason that the greatest increases should have occurred with respect to the derivatives of these secoiridoids, such as HT, its glycoside and Ty, although it is true that these increases were most noticeable in the picual variety with the greenest stages of maturity. In the case of DHPG, a general decrease was observed, and only in the arbequina variety did it increase significantly. This phenol originates from derivatives of oleuropein and verbascoside and requires treatment above 50 °C for its formation, a condition that does not occur in the virgin olive oil extraction process [23]. For the rest of the phenols, there were no significant changes after the application of the cold-pressing system, and they had concentrations in the liquid fraction exceeding those reported for riper olives in some cases [19], such as those of DHPG, verbascoside and comsegoloside.

### 4.4. Effect on the Phenolic Profile of the Oil

The results for total phenols did not indicate homogeneous behavior after the cold-pressing treatment. Although a decrease was observed in some samples, it was possible to increase the total phenols in the cases of the hojiblanca and picual varieties. Regarding the phenolic profile, it is noteworthy that the main phenols, such as oleacein and oleocanthal, hardly varied with the treatment and were within the range reported by other authors [24].

## 5. Conclusions

The application of the cold-pressing system as a stage prior to the extraction of olive oil offers a number of important advantages that could justify its use at the industrial level. It can improve the extraction of oil from greener olives, in some cases increasing their total phenol content without changing their phenolic profile, thus improving their industrial yield. Furthermore, it facilitates the subsequent separation of phases from the by-product or alperujo, increasing the concentrations of soluble sugars and some of the individual phenols, mainly HT and its glycoside. Further studies on how this treatment affects the quality parameters of virgin olive oil will be necessary to determine the true utility of this treatment at the industrial level. The results obtained for the hojiblanca sample from the following season did not indicate any effect from the cold-pressing treatment since, due to the environmental conditions of that season, lack of water caused the olives to be very dry, and this stress increased the concentration of phenols and reduced the concentration of sugars. Therefore, the most homogeneous results were obtained in the first season when irrigation and weather conditions were more normal.

The results of this study show the possible utility of the new cold-pressing technology to improve the olive oil production sector and the use of olive oil by-products. This technology needs to be tested according to the type of variety and the degree of ripeness, as it is more effective with greener samples.

## Figures and Tables

**Figure 1 antioxidants-12-01162-f001:**
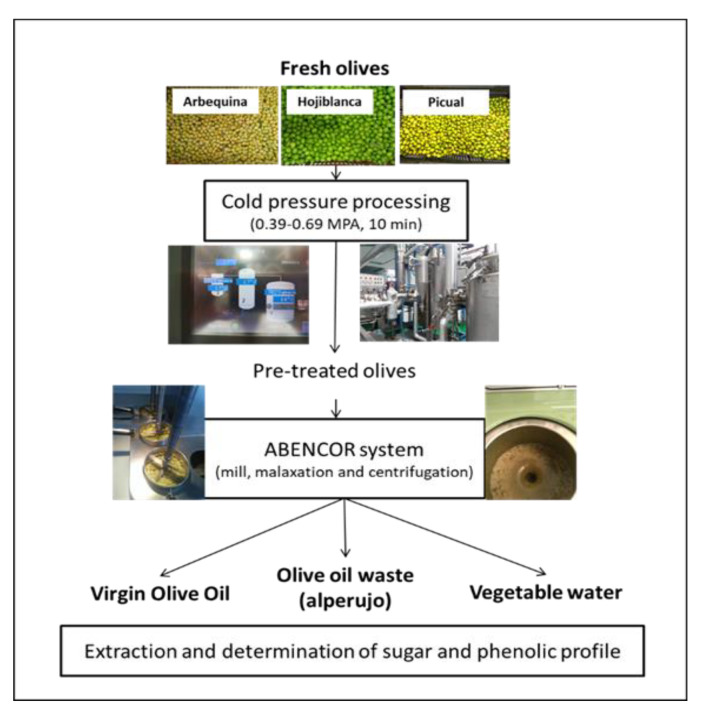
Working scheme that includes the cold-pressure treatment as a first step followed by the use of the olive oil processing system (Abencor) to obtain virgin olive oil, alperujo and vegetation water.

**Figure 2 antioxidants-12-01162-f002:**
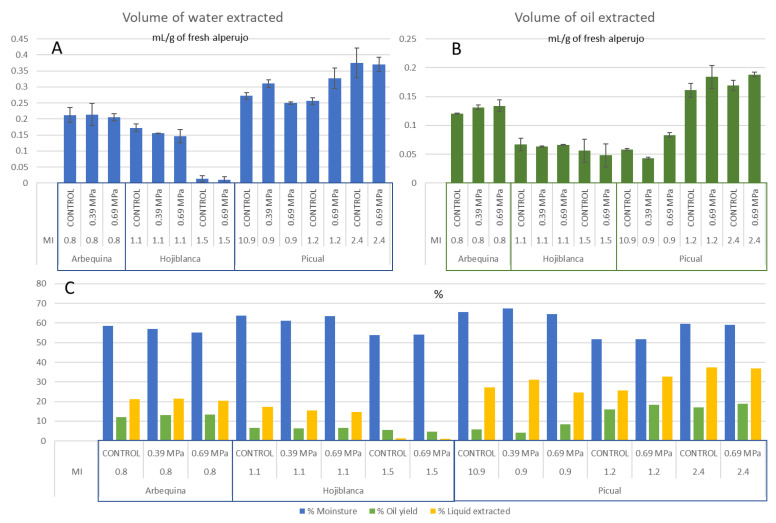
Olive variety, maturity index (MI) and treatment used for volumes of (**A**) water (mL/g of fresh alperujo) and (**B**) oil (mL/g of fresh alperujo) extracted with the Abencor system and (**C**) the percentages of moisture, oil yield and liquid extracted. All determinations were calculated in triplicate (*n* = 3).

**Table 1 antioxidants-12-01162-t001:** Sugar (g glucose eq./kg dry matter), total phenolic content (g gallic acid eq./kg dry matter) and main individual phenolics (3,4-dihydroxyphenylglycol (DHPG), hydroxytyrosol 4-β-D-glucoside (HT-Glu), hydroxytyrosol (HT), tyrosol (Ty), syringic acid (Sy A), verbascoside (Ve), luteonin-7-O-glucoside (Lu-Glu) and comsegoloside (Cg)) in the alperujo obtained from samples using different maturity indices (MIs), varieties (arbequina (Arb), hojiblanca (Ho) and picual (Pi)) and treatments (Treat).

Variety/MI		Treat	Total Sugars	Total Phenolics	Individual Phenolics (mg/kg Dry Matter)							
					DHPG	HT-Glu	HT	Ty	Sy A	Ve	Lu-Glu	Cg
Arb	0.8	Control	6.51 ± 0.30 bc *	3.14 ± 0.13 b	40.75 ± 2.04 a	593.24 ± 18.97 e	113.51 ± 8.79 a	169.97 ± 12.32 g	8.72 ± 0.84 a	345.21 ± 17.81 e	Traces	290.96 ± 10.21 c
		0.39 MPa	7.37 ± 0.13 d	3.57 ± 0.11 b	35.04 ± 1.98 a	730.41 ± 32.47 f	216.58 ± 23.41 c	166.89 ± 11.00 g	36.68 ± 2.57 c	413.13 ± 10.51 fg	Traces	403.78 ± 32.82
		0.69 MPa	7.09 ± 0.26 cd	3.47 ± 0.14 b	36.75 ± 1.87 a	755.27 ± 44.08 fg	282.09 ± 25.12 c	29.93 ± 3.80 a	7.00 ± 1.05 a	365.01 ± 19.52 e	231.35 ± 8.69 c	348.45 ± 4.90 d
Ho	1.1	Control	5.79 ± 0.32 b	2.76 ± 0.14 a	269.20 ± 11.70 h	870.91 ± 33.75 g	141.02 ± 5.64 a	89.34 ± 4.46 e	31.08 ± 3.44 c	446.60 ± 15.50 g	42.92 ± 7.78 b	188.51 ± 15.30 a
		0.39 MPa	6.69 ± 0.21 c	3.31 ± 0.15 b	190.38 ± 9.51 f	1585.50 ± 45.60 h	386.80 ± 30.01 d	127.04 ± 12.27	28.87 ± 1.20 c	795.60 ± 20.67 h	477.77 ± 23.15 d	179.52 ± 7.84 a
		0.69 MPa	6.75 ± 0.29 c	3.32 ± 0.16 b	56.24 ± 4.22 b	1669.91 ± 52.67 h	469.14 ± 26.80 de	112.38 ± 17.12 f	48.00 ± 2.77 d	902.52 ± 32.60 i	657.76 ± 30.50 e	228.23 ± 6.23 b
	1.5	Control	3.67 ± 0.06 a	12.24 ± 0.07 e	87.72 ± 7.24 c	208.7 ± 10.12 b	308.69 ± 9.04 c	73.18 ± 5.60 e	Traces	Traces	1124.78 ± 33.81 f	841.26 ± 27.85 e
		0.69 MPa	3.47 ± 0.05 a	11.77 ± 0.03 de	89.86 ± 5.50 c	204.27 ± 12.29 b	186.48 ± 14.77 b	99.72 ± 3.42 e	Traces	Traces	1084.01 ± 42.55 f	934.92 ± 32.40 e
Pi	0.9	Control	5.08 ± 0.02 b	2.40 ± 0.08 a	70.75 ± 2.87 c	364.42 ± 12.03 cd	130.30 ± 9.68 a	60.66 ± 3.30 c	Traces	Traces	Traces	259.43 ± 10.45 bc
		0.39 MPa	4.40 ± 0.42 ab	2.30 ± 0.09 a	61.16 ± 2.11 b	348.13 ± 7.42 c	150.21 ± 18.87 ab	61.74 ± 4.02 c	Traces	Traces	Traces	166.79 ± 4.35 a
		0.69 MPa	5.04 ± 0.04 b	2.38 ± 0.05 a	102.41 ± 1.20 d	450.61 ± 21.80 d	138.13 ± 12.43 a	67.48 ± 7.60 cd	Traces	Traces	Traces	324.54 ± 15.60 d
	1.2	Control	20.05 ± 0.24 f	9.88 ± 0.12 d	176.85 ± 11.87 ef	85.98 ± 16.12 a	256.53 ± 27.58 c	43.65 ± 2.32 b	9.36 ± 1.29 a	101.14 ± 3.20 d	12.18 ± 1.10 a	284.37 ± 16.27 c
		0.69 MPa	20.35 ± 0.05 f	10.03 ± 0.02 d	152.79 ± 9.65 e	237.99 ± 30.09 b	532.20 ± 18.00 e	71.00 ± 4.33 d	12.79 ± 2.00 b	30.14 ± 1.58 a	201.87 ± 22.37 c	284.24 ± 12.77 c
	2.4	Control	10.51 ± 0.10 e	5.14 ± 0.25 c	340.12 ± 12.77 i	342.80 ± 29.36 bc	1040.91 ± 48.10 f	117.38 ± 9.23 f	11.72 ± 1.78 ab	47.17 ± 3.24 c	191.26 ± 18.20 c	324.33 ± 16.80 d
		0.69 MPa	9.16 ± 0.38 e	4.47 ± 0.19 c	200.20 ± 15.03 fg	401.57 ± 26.48 d	1047.75 ± 52.20 f	111.15 ± 7.74 f	12.23 ± 1.92 b	37.18 ± 1.01 b	222.21 ± 20.13 c	290.22 ± 17.62 cd

* Means with the same letter in the same row were not significantly different, *p* < 0.05. Traces, < 0.01 mg/L. All determinations were calculated in triplicate (*n* = 3).

**Table 2 antioxidants-12-01162-t002:** Sugar (g glucose eq./kg dry matter), total phenolic content (g gallic acid eq./kg dry matter) and main individual phenolics (3,4-dihydroxyphenylglycol (DHPG), hydroxytyrosol 4-β-D-glucoside (HT-Glu), hydroxytyrosol (HT), tyrosol (Ty), syringic acid (Sy A), verbascoside (Ve), p-coumaric acid (p-Co) and comsegoloside (Cg)) in the water fractions obtained from samples using different maturity indices (MIs), varieties (arbequina (Arb), hojiblanca (Ho) and picual (Pi)) and treatments (Treat).

Variety/MI		Treat	Total Sugars	Total Phenolics	Individual Phenolics (mg/L)							
					DHPG	HT-Glu	HT	Ty	Sy A	Ve	p-Co	Cg
Arb	0.8	Control	28.12 ± 0.34 b *	5.22 ± 0.19 d	18.23 ± 1.8 a	300.31 ± 17.08 c	184.26 ± 14.21 b	123.25 ± 2.99 d	15.79 ± 1.39 a	192.89 ± 11.07 d	6.51 ± 0.38 a	219.10 ± 4.73 c
		0.39 MPa	31.11 ± 0.61 b	5.15 ± 0.24 d	18.99 ± 1.92 a	271.29 ± 13.20 c	174.19 ± 15.85 b	122.74 ± 6.06 d	15.97 ± 1.54 a	212.02 ± 17.58 d	6.25 ± 0.08 a	224.20 ± 9.77 c
		0.69 MPa	26.96 ± 0.48 b	5.42 ± 0.25 d	16.74 ± 1.80 a	295.74 ± 25.53 c	175.45 ± 19.23 b	43.46 ± 2.42 a	17.53 ± 2.89 ab	190.67 ± 17.77 d	4.91 ± 0.64 a	212.15 ± 0.54 c
Ho	1.1	Control	35.27 ± 0.33 c	4.97 ± 0.23 cd	95.28 ± 13.12 c	552.89 ± 20.01 d	196.57 ± 22.48 bc	118.58 ± 6.97 d	12.71 ± 2.76 a	263.95 ± 16.24 e	17.26 ± 1.09 c	95.01 ± 5.73 a
		0.39 MPa	36.29 ± 1.14 c	5.22 ± 0.15 d	120.29 ± 16.39 d	550.66 ± 7.00 d	218.76 ± 13.91 c	121.27 ± 5.99 d	15.20 ± 2.96 a	277.84 ± 18.96 e	17.71 ± 1.46 c	102.86 ± 7.10 a
		0.69 MPa	36.28 ± 0.67 c	5.29 ± 0.20 d	137.91 ± 9.99 de	559.17 ± 17.84 d	202.86 ± 2.61 c	123.85 ± 9.93 d	20.79 ± 1.58 b	247.80 ± 13.69 e	18.20 ± 1.51 c	100.75 ± 6.16 a
	1.5	Control	17.80 ± 1.22 a	6.79 ± 0.66 e	34.21 ± 3.90 b	155.82 ± 7.42 b	558.40 e ±12.24	62.15 ±3.08 b	Traces	Traces	12.87 ± 0.79 b	799.05 ± 22.75 d
		0.69 MPa	15.42 ± 0.55 a	5.95 ± 0.78 de	32.02 ± 4.14 b	148.24 ± 8.80 b	520.63 ± 15.61 e	44.08 ± 2.20 a	Traces	Traces	11.92 ± 1.05 b	805.12 ± 27.40 d
Pi	0.9	Control	33.12 ± 0.92 bc	4.72 ± 0.08 c	198.62 ± 6.10 f	126.41 ± 3.47 a	141.34 ± 2.60 a	76.51 ± 1.11 b	18.14 ± 1.65 b	79.02 ± 9.88 bc	13.67 ± 0.25 b	190.01 ± 3.01 b
		0.39 MPa	28.90 ± 1.23 b	4.48 ± 0.21 c	164.81 ± 24.30 e	160.40 ± 4.30 b	180.31 ± 18.07 b	86.62 ± 4.66 c	16.91 ± 1.19 ab	85.15 ± 5.58 c	14.10 ± 0.94 b	185.93 ± 9.59 b
		0.69 MPa	32.00 ± 0.36 bc	4.65 ± 0.22 cd	207.21 ± 18.62 f	147.11 ± 5.71 b	160.19 ± 17.41 ab	90.32 ± 4.32 c	19.09 ± 0.74 b	90.61 ± 5.16 c	13.83 ± 0.62 b	188.66 ± 8.38 b
	1.2	Control	32.63 ± 0.22 bc	3.42 ± 0.04 b	204.21 ± 12.35 f	590.01 ± 24.09 d	438.11 ± 19.18 d	126.71 ± 6.81 d	25.76 ± 1.12 c	72.50 ± 2.07 b	34.78 ± 0.87 d	597.66 ± 24.50d
		0.69 MPa	34.38 ± 0.91 c	3.50 ± 0.08 b	99.70 ± 11.47 b	302.07 ± 21.20 c	798.12 ± 17.50 f	112.33 ± 9.80 cd	25.84 ± 2.55 c	56.42 ± 1.08 a	31.72 ± 1.29 d	602.12 ± 32.00 d
	2.4	Control	32.05 ± 0.06 c	3.00 ± 0.06 a	113.50 ± 7.11 bc	304.92 ± 15.29 c	842.51 ± 20.03 f	148.77 ± 7.88 de	27.05 ± 3.42 c	54.01 ± 2.22 a	30.87 ± 2.07 d	603.50 ± 27.09 d
		0.69 MPa	33.85 ± 0.06 c	2.88 ± 0.11 a	171.86 ± 5.42 d	267.97 ± 9.88 c	520.97 ± 23.77 e	153.22 ± 5.43 e	28.97 ± 2.20 c	70.41 ± 3.91 b	35.86 ± 2.57 d	641.07 ± 37.84 d

* Means with the same letter in the same row were not significantly different, *p* < 0.05. All determinations were calculated in triplicate (*n* = 3).

**Table 3 antioxidants-12-01162-t003:** Total phenolic content (g gallic acid/kg fresh alperujo) and main individual phenolics (3,4-dihydroxyphenylglycol (DHPG), hydroxytyrosol (HT), tyrosol (Ty), vanillic acid, oleocanthal and oleacein) for the olive oils obtained from samples using different maturity indices (MIs), varieties (arbequina (Arb), hojiblanca (Ho) and picual (Pi)) and treatments (Treat).

Variety/MI		Treat	Total Phenolics	Individual Phenols (mg/L)					
				DHPG	HT	Ty	Vanillic Acid	Oleocanthal	Oleacein
Arb	0.8	Control	0.8 ± 0.0	1.99 ± 0.37 ef*	11.01 ± 1.16 c	4.92 ± 0.81 ab	6,46 ± 1.03 g	311,23 ± 12.98 f	177,53 ± 11.09 bc
		0.39 MPa	0.6 ± 0.0	0.68 ± 0.26 c	8.11 ± 0.67 bc	3.50 ± 0.04 a	4,73 ± 0.86 f	243,95 ± 11.23 e	140,96 ± 12.08 a
		0.69 MPa	0.6 ± 0.0	1.46 ± 0.27 e	8.82 ± 0.45 c	4.93 ± 0.07 b	5,41 ± 0.71 fg	233,47 ± 15.12 e	141,98 ± 6.66 a
Ho	1.1	Control	0.9 ± 0.0	0.36 ± 0.07 ab	10.27 ± 0.11 c	7.01 ± 0.06 d	4,78 ± 0.70 f	222,73 ± 9.88 e	193,77 ± 7.01 c
		0.39 MPa	0.6 ± 0.0	0.60 ± 0.11 c	8.31 ± 1.60 bc	5.24 ± 0.09 b	3,58 ± 0.63 ef	189,84 ± 10.42 d	163,06 ± 9.89 ab
		0.69 MPa	1.1 ± 0.0	0.30 ± 0.07 a	9.80 ± 1.42 c	6.36 ± 0.10 c	1,05 ± 0.08 a	215,20 ± 15.18 de	170,67 ± 7.09 b
	1.5	Control	1.3 ± 0.0	nd	12.5 ± 2.31 c	4.3 ± 1.05 a	nd	222.9 ± 17.18 e	270.9 ±8.50 d
		0.69 MPa	1.2 ± 0.0	nd	11.9 ± 1.99 c	3.8 ± 1.11 a	nd	145.6 ± 12.33 b	252.8 ± 13.2 d
Pi	0.9	Control	0.5 ± 0.0	1.33 ± 0.11 de	6.18 ± 0.01 a	3.46 ± 0.21 a	2,98 ± 0.08 e	140,62 ± 7.02 b	166,52 ± 7.77 ab
		0.39 MPa	0.7 ± 0.0	0.40 ± 0.01 b	28.30 ± 1.03 d	8.30 ± 0.41 d	2,90 ± 0.07 e	51,47 ± 5.63 a	167,08 ± 9.02 ab
		0.69 MPa	0.5 ± 0.0	1.20 ± 0.21 d	7.91 ± 0.52 bc	9.37 ± 0.56 de	2,53 ± 0.11 d	162,24 ± 4.32 c	169,60 ± 10.10 ab
	1.2	Control	1.2 ± 0.0	2.14 ± 0.04 f	33.14 ± 2.40 d	10,87 ± 1.03 e	1.73 ± 0.02 c	146,04 ± 7.77 b	392,76 ± 15.50 e
		0.69 MPa	1.0 ± 0.0	0.40 ± 0.01 b	8.70 ± 0.22 bc	13,90 ±1.00 f	1.23 ± 0.01 a	134,07 ± 6.99 b	230,77 ± 10.9 d
	2.4	Control	1.2 ± 0.0	1.90 ± 0.02 ef	73.15 ± 2.47 f	17,88 ± 2.06 g	1.55 ± 0.01 b	146,66 ± 11.70 b	491,54 ± 22.70 f
		0.69 MPa	1.5 ± 0.0	4.30 ± 0.65 g	57.60 ± 4.84 e	16,61 ± 1.93 a	1.23 ± 0.01 a	140,02 ± 10.02 b	434,80 ± 17.95 ef

* Means with the same letter in the same row were not significantly different, *p* < 0.05. nd: not detected. All determinations were calculated in triplicate (*n* = 3).

## Data Availability

The data are contained within the article.

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
