# Peer review of "Application of a Cold-Pressing Treatment to Improve Virgin Olive Oil Production and the Antioxidant Phenolic Profile of Its by-Products"

_antioxidants, 2023, doi:10.3390/antiox12061162_

Round 1
Reviewer 1 Report
In this manuscript the authors describe the advantages of applying the cold pressing system in the preliminary stages of oil extraction. The advantages described are in the better extraction of oil, in green olives, with an increase in total phenols and facilitates the separation of the phases from the by-product with an increase in sugars and individual phenols. The article is well structured and clearly written in good English. However, the results are based on a small number of olive cultivars, a limited number of observations for each of them in the various treatments and in different ripening stages. This would certainly allow for a greater solidity of the results obtained.
For this reason, I think I have to ask the authors for an effort and include further experiments.
Thank you.
Author Response
Reviewer 1:
In this manuscript the authors describe the advantages of applying the cold pressing system in the preliminary stages of oil extraction. The advantages described are in the better extraction of oil, in green olives, with an increase in total phenols and facilitates the separation of the phases from the by-product with an increase in sugars and individual phenols. The article is well structured and clearly written in good English. However, the results are based on a small number of olive cultivars, a limited number of observations for each of them in the various treatments and in different ripening stages. This would certainly allow for a greater solidity of the results obtained.
Response: The trial started in the season in which COVI was initiated, so the samples that were obtained were either very mature or deteriorated, making the results unreliable. The 2020/2021 season was a normal season and is where most samples were processed. In the following season, there was a severe drought that has lasted into the current season, and the samples had very low humidity, which made oil extraction very difficult. Despite this, one of the samples has been included in the hojiblanca article and its low humidity and results have been justified in the text. In spite of this, the number of samples has been triplicated and, as it is a preliminary study, the data obtained are reliable and reproducible for different varieties, thus completing and supporting the conclusions of the work.
Reviewer 2 Report
The article presents interesting results about quality of olive oil influenced by technological parameters - modified process. The study was conducted using 3 different olive varieties as well as different stages of maturation what was important.
The results are presented in table format and it would be interesting to show in general the effect of used treatment on total polyphenols contents ect. in the figure format.
Please use in all text 'moisture' not 'humidity'.
Please add under the table the number of samples: n=
'Vegetable water' is not the best name;
later on (L 335) is also 'vegetation water'???
The English should be check carefully without all aticle as there are many mistakes as:
L 138 - additional 'of' which should be removed
L 145 - should be 'works' not 'woks'
The English should be checked very carefully
Author Response
Reviewer 2:
The article presents interesting results about quality of olive oil influenced by technological parameters - modified process. The study was conducted using 3 different olive varieties as well as different stages of maturation what was important.
The results are presented in table format and it would be interesting to show in general the effect of used treatment on total polyphenols contents ect. in the figure format.
Response: Table 1 concerning moisture content and oil and water extraction yields has been converted into a figure to make it more visual. In the case of phenols, it has been kept in tables because of the complexity of the data and so that the total and individual contents could be discussed at the same time.
Please use in all text 'moisture' not 'humidity'.
Response: It has been corrected in the text.
Please add under the table the number of samples: n=
Response: It has been inserted in each table and in figure 2.
'Vegetable water' is not the best name;
later on (L 335) is also 'vegetation water'???
Response: The correct name is vegetation water. It has been correct in the text.
The English should be check carefully without all aticle as there are many mistakes as:
L 138 - additional 'of' which should be removed
L 145 - should be 'works' not 'woks'
Response: The text has been carefully checked and corrected.
Round 2
Reviewer 1 Report
Dear Authors,
thank you for the effort you have made to improve your manuscript which has certainly undergone a great improvement.
Unfortunately, the pandemic and the "unfavorable" climate of recent years have certainly not facilitated research activities either. I understand and therefore appreciate the effort made. However, if the project and the results have been affected, turning to a journal with an adequate impact factor certainly does not diminish the importance of the research.
Please correct line 99 “MADURATION” with maturation.
After that I consider that the article can be accepted.
Thank you again for your work,
Reviewer 2 Report
The present version of article was improved and can be accepted.
Please correct in the text "humidity" on 'moisture' as it is mixed.
English was improved